# Increased Precipitation Shapes Relationship between Biochemical and Functional Traits of *Stipa glareosa* in Grass-Dominated Rather than Shrub-Dominated Community in a Desert Steppe

**DOI:** 10.3390/plants9111463

**Published:** 2020-10-29

**Authors:** Ya Hu, Xiaoan Zuo, Ping Yue, Shenglong Zhao, Xinxin Guo, Xiangyun Li, Eduardo Medina-Roldán

**Affiliations:** 1Urat Desert-Grassland Research Station, Northwest Institute of Eco-Environment and Resources, Chinese Academy of Science, Lanzhou 730000, China; huya@lzb.ac.cn (Y.H.); yueping@lzb.ac.cn (P.Y.); zhaoshl@lzb.ac.cn (S.Z.); guoxinxin@lzb.ac.cn (X.G.); lixiangyun19@mails.ucas.ac.cn (X.L.); 2Naiman Desertification Research Station, Northwest Institute of Eco-Environment and Resources, Chinese Academy of Science, Lanzhou 730000, China; 3University of Chinese Academy of Sciences, Beijing 100049, China; 4Department of Health and Environmental Science, Xi’an Jiaotong-Liverpool University, Suzhou 215000, China; Eduardo.Medina-Roldan@xjtlu.edu.cn

**Keywords:** precipitation gradient, climate change, functional trait, biochemical trait, desert steppe

## Abstract

Understanding the effects of precipitation variations on plant biochemical and functional traits is crucial to predict plant adaptation to future climate changes. The dominant species, *Stipa glareosa*, plays an important role in maintaining the structure and function of plant communities in the desert steppe, Inner Mongolia. However, little is known about how altered precipitation affects biochemical and functional traits of *S. glareosa* in different communities in the desert steppe. Here, we examined the responses of biochemical and functional traits of *S. glareosa* in shrub- and grass-dominated communities to experimentally increased precipitation (control, +20%, +40%, and +60%). We found that +40% and +60% increased plant height and leaf dry matter content (LDMC) and decreased specific leaf area (SLA) of *S. glareosa* in grass community. For biochemical traits in grass community, +60% decreased the contents of protein and chlorophyll b (Cb), while +40% increased the relative electrical conductivity and superoxide dismutase. Additionally, +20% increased LDMC and malondialaenyde, and decreased SLA and protein in shrub community. Chlorophyll a, Cb, carotenoids, protein and superoxide dismutase in the grass community differed with shrub community, while +60% caused differences in SLA, LDMC, leaf carbon content, malondialaenyde and peroxidase between two communities. The positive or negative linear patterns were observed between different functional and biochemical traits in grass- rather than shrub-community. Soil water content explained changes in some biochemical traits in the grass community, but not for functional traits. These results suggest that increased precipitation can affect functional traits of *S. glareosa* in the grass community by altering biochemical traits caused by soil water content. The biochemical and functional traits of *S. glareosa* were more sensitive to extreme precipitation in grass- than shrub-community in the desert steppe. Our study highlights the important differences in adaptive strategies of *S. glareosa* in different plant communities at the same site to precipitation changes.

## 1. Introduction

Global climate changes are increasingly altering precipitation patterns in different regions [1]. Some climate change models have predicted that the future precipitation will increase in arid areas [2,3], which further affects community structure and assembly. In view of the predictive power of functional traits in plant responses to precipitation changes in grasslands [4], the studies based on functional traits can provide an insight into plant adaptive strategies along the precipitation gradient [5]. Increasing precipitation increases plant height, specific leaf area (SLA), leaf carbon content (LCC) and leaf nitrogen content (LNC), while decreasing leaf dry matter content (LDMC) and leaf thickness, suggesting that plants can adapt to precipitation changes by altering the specific traits [4,6,7]. However, these results are usually obtained from relatively humid areas. Plants with smaller SLA and thicker leaves show the higher water use efficiency in arid and semi-arid areas [8]. Plants with the higher LNC are competitive in the lower precipitation conditions [9]. Therefore, there is a growing need to study how plant functional traits respond to increased precipitation in drylands, which is helpful to understand plant adaption to future precipitation changes.

Numerous studies have shown that short-term environmental changes can induce rapid biochemical responses of herbaceous plants [10,11,12]. Plant physiological and biochemical processes, such as photosynthesis, osmoregulation and antioxidation play important roles in mediating local adaptation to abiotic factors [13]. The differences in biochemical tolerance and competitive ability contribute to the distribution of species along a precipitation gradient [14]. The mechanism of plant resistance to drought by adjusting the content and proportion of photosynthetic pigment, increasing the content of osmotic substance and activity of antioxidant enzymes, has been extensively studied [10,15,16]. Growing evidence has shown that the responses of plants to drought stress and extreme precipitation can be regulated by some common pathways [17]. Indeed, flooding or extreme precipitation decrease the contents of chlorophyll and carotenoids (Cx) and further affect absorption, transmission and conversion of light energy [18]. Changes in osmoregulation substance (e.g., proline and protein) content can maintain the cell water potential at a relatively stable level to adapt to altered precipitation [19]. The increase in antioxidant enzyme (superoxide dismutase (SOD) and peroxidase (POD)) activity is beneficial for removing reactive oxygen species, reducing membrane lipid peroxidation products, malondialaenyde (MDA), and maintaining the membrane permeability of plants under water stress [16,20,21]. Similar to functional traits, using the biochemical traits can help to explore plant adaption strategies to future precipitation changes. However, few attempts have been made to investigate plant biochemical responses to increased precipitation in the manipulative field experiment.

The functional and biochemical traits can exhibit the rapid or transient responses of plants to climate changes [4,12]. The responses of plant functional traits in grassland communities along a resource gradient are based on the biochemical tradeoffs between a series of related traits [22]. Some studies have documented that using an integrated approach of plant functional and biochemical traits can better explore the species-specific responses to drought and nutrient deficiency [10,11]. Plant community structure and function in grasslands are more susceptible to the responses of dominant plants to precipitation changes [23,24]. However, there are few studies on the specific responses of dominant species to increased precipitation in the field grassland experiment by a combination of functional and biochemical traits.

The Inner Mongolia desert steppe is located in the transition zone between desert and steppe, in which it develops two kinds of plant communities dominated by shrub and grass [25]. Shrub encroachment is also a worldwide phenomenon in this region, thus leading to the conversion from grassland to desert shrubland [26]. Our previous studies have shown that changes in species richness and above biomass in grass-dominated community are consistent with precipitation changes, but the relationships between these two and precipitation are nonlinear in the shrub community [25]. In addition to the differences of community characteristics, the same plant has different performance in different communities. For instance, manipulated neighbor shrubs have significant effects on growth and biochemistry of *Stipa tussocks* [27]. However, it is still unclear whether the functional and biochemical traits of the common species in different communities have different ecological adaptability. *Stipa glareosa*, a dominant plant in the desert steppe in Inner Mongolia, can coexist in both grass and shrub communities. To explore the responses of *S. glareosa* to increased precipitation in grass and shrub communities at one site is crucial for understanding plant adaptive strategies in the desert steppe.

In this study, we investigated how experimentally increased precipitation affected plant functional and biochemical traits of *S. glareosa* in two adjacent plant communities in the desert steppe in Inner Mongolia. We hypothesized that (1) the responses of the most of functional and biochemical traits of *S. glareosa* in grass community to increased precipitation differed with shrub community due to shrub encroachment effects; (2) changes in functional traits are based on biochemical processes in grasslands, thus some biochemical traits were related to functional traits in grass community; (3) soil water content induced by altered precipitation could explain some functional and biochemical traits of *S. glareosa* in the desert steppe.

## 2. Results

### 2.1. Effects of Increased Precipitation on Functional Traits

Precipitation had a significant effect on plant height, and the SLA and LDMC of *S. glareosa*. The community significantly affected SLA, LDMC and LCC, and the interaction between precipitation and community was significant for leaf thickness and LCC (*p* < 0.05, Appendix A). We found that 40% and 60% increases in precipitation increased plant height of *S. glareosa* in the grass community, but there were no significant effects on plant height of *S. glareosa* in the shrub community (*p* > 0.05, Figure 1a). Increases of 40% and 60% precipitation reduced SLA and increased LDMC of *S. glareosa* in the grass community. A 20% increased precipitation decreased SLA and increased LDMC of *S. glareosa* in the shrub community (Figure 1c,d). Leaf thickness, LCC and LNC had no significant responses to increased precipitation (*p* > 0.05, Figure 1b,e,f).

### 2.2. Effects of Increased Precipitation on Biochemical Traits

Precipitation had a significant effect on Cb, protein, MDA content, relative electrical conductivity (REC), SOD and POD activity. The community significantly affected Ca, Cb, Cx, protein, MDA content, REC, SOD and POD activity, and the interaction between precipitation and community was significant for Ca, Cx, proline and protein contents (*p* < 0.05, Appendix A). The pigment content of *S. glareosa* differed between two communities, and +60% reduced Cb content in grass community (Figure 2a–c). Increased precipitation and community had no significant effects on proline content (*p* > 0.05, Figure 2d). Protein content was reduced with the precipitation increase in shrub community and there were significant differences between two communities (Figure 2e). A 40% increase in precipitation increased MDA content in the grass community and a 20% increase lead to increased MDA content in the shrub community (Figure 2f). A 40% increase in precipitation increased REC in the grass community and the SOD activity in both of the two communities (Figure 2g,h). A 20% increase in precipitation significantly increased the POD activity of *S. glareosa* in the grass community, and there were no differences in the shrub community (Figure 2i).

### 2.3. Correlations between Functional and Biochemical Traits

Correlation analyses among six functional and nine biochemical traits are presented in Figure 3. LDMC was negatively correlated with Ca, Cx and proline content and positively correlated with MDA content of *S. glareosa* in grass community (*p* < 0.05). Proline content was negatively related to plant height (*p* < 0.05) and protein content was positively correlated with leaf thickness in the grass community (*p* < 0.01). MDA content and SOD activity had a negative correlation with SLA in the grass community (*p* < 0.01). REC and SOD activity were negatively correlated with LCC in the grass community (*p* < 0.05). Proline content and SOD activity had a positive correlation with LNC of *S. glareosa* in the shrub community.

### 2.4. Correlations between Soil Water Content and Biochemical Traits

Increases of 40% and 60% in precipitation increased soil water content in the shrub and grass communities, respectively, and soil water content in the shrub community was higher than in the grass community (Appendix A). Soil water content was negatively correlated with Ca, Cb and Cx contents and was negatively correlated with the proline content of *S. glareosa* in the grass community (*p* < 0.05, Figure 4). In addition, soil water content was negatively correlated with protein content and positively correlated with SOD activity of *S. glareosa* in the shrub community (*p* < 0.05). However, there were no correlations between soil water content and functional traits (*p* > 0.05, Appendix A).

## 3. Discussion

For functional traits, we found that *S. glareosa* responded to increased precipitation by changing plant height, SLA and LDMC in the grass community, which supports that the SLA and LDMC are two key functional traits reflecting plant adaption, because they are more sensitive to environmental changes than other functional traits [28,29]. Increases of 40% and 60% in precipitation produced taller *S. glareosa* plants in the grass community, which is consistent with other studies, showing that high precipitation increases the availability of soil nutrients and water content, thus further accelerating plant growth in the grassland [4,30]. High precipitation decreased SLA and increased the LDMC of *S. glareosa* in the grass community, which is similar to the findings that native grasses display low SLA with high precipitation in drylands [4]. However, these results are opposite to the studies on changes in SLA and LDMC along the natural precipitation gradient [3,7]. This can be interpreted by the unique morphology and anatomy of *Stipa* or species-specific strategy of resource acquisition [31]. *Stipa* leaves are completely rolled up into needle shapes to lessen the exposed area to the air and a thick cuticle covers the outer surface to make it resistant to moisture diffusion. These structural features could contribute to the adaption of *Stipa* to environment changes. The decreased SLA and increased LDMC of *S. glareosa* under increased precipitation can reflect the more conservative resource strategies of plants for storing more water in tissues in arid environments [7,32]. Thus, the responses of functional traits to increased precipitation suggest a trade-off between the resource acquisition and conservation of *S. glareosa*, which seems to be crucial for species adaption in arid areas.

For biochemical traits, we observed that Cb and protein contents of *S. glareosa* tended to decrease with increased precipitation, which is consistent with the dilution effect that increased water content reduces the substance contents per unit mass. A 40% increase in precipitation increased the MDA content of *S. glareosa* in the grass community, suggesting that the lipid peroxidation induced by reactive oxygen species has led to cellular oxidative damage [16,21]. Similarly, increases of 20% and 40% in precipitation increased REC of *S. glareosa* in the grass community, which indicates an increase in membrane permeability [33]. However, plants could resist adverse effects by regulating the antioxidant enzyme system. As mentioned, the increases in POD and SOD activities under +20% and +40% treatments, are the positive responses for *S. glareosa* to eliminate reactive oxygen species and alleviate the cell membrane damage [16,34]. The differences in functional and biochemical trait responses of *S. glareosa* between grass community and shrub community support our first hypothesis. Moreover, the linear regression analysis results differed between grass and shrub community.

SLA and LDMC were significantly correlated to Ca, Cb, proline, MDA contents and SOD activity in grass community. This agrees with other studies that the biochemical traits are directly related to leaf functional traits [35]. Our results also suggest that a close association between functional and biochemical traits of plants is shaped by altered precipitation, fully supporting the second hypothesis.

In addition, we observed that the soil water content could well explain some biochemical traits, but not for functional traits, partly supporting the third hypothesis. Probably, the dominant *S. glareosa* in the grass community can alter some key biochemical traits in order to adapt to high precipitation. According to Sherrard et al. [15], who reported that environmental differences in soil water content affected some plant biochemical traits rather than functional traits, the relative higher soil water content in shrub community is mainly due to the “fertility islands” formed by shrub establishments in grasslands which can enhance soil nutrients and water under shrubs, as well as decrease the temperature and light intensity, thus reducing water loss [36]. Shrubs, having the high resource acquisition ability, present the strong competition to soil water in the shrub encroaching community [37,38], probably leading to the weak associations between most biochemical traits of *S. glareosa* with soil water content in the shrub community. These results suggest that *S. glareosa* in different communities displays the different adaptive strategies to increased precipitation by modulating some biochemical and functional traits. This may be explained by the specific ecological niche of *S. glareosa* in different communities [39,40].

## 4. Materials and Methods

### 4.1. Study Area

The study area was located in Urat Rear Banner, western Inner Mongolia, China (41°25′ N, 106°58′ E, 1650 m a.s.l.) (Appendix A). It had a continental arid climate with cold and long winters, short and cool summers, scarce precipitation and strong wind. The average annual temperature was 3.9 °C, average annual precipitation was 180 mm, average annual wind speed was 5 m s^−1^. Vegetation in this area had a high mosaic distribution of shrub- and grass-dominated communities. The experimental plots were placed in the long-term experimental observation site of the Urat Desert Grassland Research Station, Northwest Institute of Eco-Environment and Resources, Chinese Academy of Sciences. We set two separated experiments in a grass community and a shrub community with different community characteristics (Appendix A). Dominant plant species in the grass community were *S. glareosa*, *Peganum harmala*, *Allium polyrhizum*, *Salsola collina*, *Allium mongolicum*, *Asparagus cochinchinensi*, *Artemisia frigida* and *Oxytropis platysema*. The dominant plant species in the shrub community were *Reaumuria songarica*, *Salsola collina*, *Salsola passerina*, *S. glareosa*, *Allium polyrhizum*, *Allium mongolicum*, *Convolvulus ammannii* and *Ajania fruticulosa*. Soil types in the study area were identified as the brown soil and grey brown desert soil [41].

### 4.2. Experimental Design

The grass- and shrub-communities in the desert steppe were distributed at the landscape scale from south to north. Two coexisting plant communities within the same location with approximately 400 m apart intervals were selected as experimental areas. We applied a durable apparatus to control precipitation for long-term monitoring of vegetation and soil traits (Appendix A). This instrument intercepted 20%, 40% and 60% of the rainfall with V-shaped polycarbonate plastic strips, respectively, into the collection tank that then flowed into the plot through the distributary dropper to conduct both decreased and increased precipitation [41]. To avoid mutual interference, a buffer zone of 2 m was set aside for each plot. We selected 24 plots, each 4 m × 4 m, with 3 increased precipitation treatments—+60%, +40%, +20% and CK (natural precipitation) (Appendix A)—to investigate the effect of increased precipitation on functional and biochemical traits of *S. glareosa*. The precipitation was obtained by the weather station of the experimental station. The experimental site has been enclosed since 2010 and all treatments have lasted three years since 2015.

### 4.3. Determination of Traits

We selected six functional traits related to plant growth, resource use strategies and ecosystem function [42,43] and nine biochemical traits related to photosynthesis, osmoregulation and antioxidation to detect the stress resistance and adaption of plants to precipitation changes (Appendix A). In August of 2018, the fast-growing stage of *S. glareosa*, we measured plant height of three *S. glareosa* with a steel tape in each experimental plot. Several matured and healthy leaves of *S. glareosa* were selected in the morning for measuring leaf functional and biochemical traits. The leaf thickness, SLA and LDMC were measured according to standard protocols [42]. The LCC and LNC were determined by elemental analyzer (Costech, Milano, Italy). In addition, Ca, Cb and Cx contents were quantified according to the absorption spectra method [44]. Proline content was measured by the ninhydrin colorimetry method [19]. Protein content was determined by the Coomassie brilliant blue G-250 method [45]. MDA content was measured by thiobarbituric acid colorimetry method [46]. REC was tested by conductance meter (Sanxin, Shanghai, China) [47]. SOD activity was analyzed with the nitroblue tetrazolium staining method [48]. POD activity was determined by the guaiac wood phenol method [34].

### 4.4. Determination of Soil Water Content

At the same time, we measured the soil water content by oven-drying method for a 0 to 10 cm soil layer. We collected the soil samples with a soil drill and oven dried the samples at 105 °C for 6–8 h to constant weight. Soil water content is calculated as following:$$\mathrm{Soil}\;\mathrm{water}\;\mathrm{content}=\left(\mathrm{Ww}-\mathrm{Wd}\right)/\left(\mathrm{Ww}-\mathrm{Wb}\right)\times 100\%$$
where Ww is the weight for empty soil box and wet soil (g), Wd is the empty soil box weight and dry soil (g), Wb is the weight for empty soil box (g).

### 4.5. Statistical Analysis

All data are presented as the mean ± standard error. The two-way analysis of variance (ANOVA) was used to analyze the effects of precipitation, community and their interaction on leaf functional and biochemical traits. One-way ANOVA was used to examine separate effects of precipitation and community on functional and biochemical traits and multiple comparisons were tested by Least-significant difference (LSD, *p* < 0.05). The linear regression analysis was used to determine the relationships between functional and biochemical traits of *S. glareosa* as well as traits and soil water content. Statistical analyses and plots were implemented using SPSS 22.0 and SigmaPlot 12.5, respectively.

## 5. Conclusions

This study highlights that *S. glareosa* in grass and shrub communities displays different trait-based adaption mechanisms to precipitation changes. Extreme precipitation could alter functional traits (plant height, SLA and LDMC) and biochemical traits (Cb, protein content, REC and SOD) of *S. glareosa* in the grass community, while four functional and biochemical traits in the shrub community responded to increased precipitation with low intensity. These results suggest that shrub encroachment can affect the response or adaption of *S. glareosa* to precipitation changes. The effects of increased precipitation in different intensities on functional and biochemical traits of *S. glareosa* are dependent on shrub or grass community types. Some key biochemical traits of *S. glareosa* in the grass community resulted from soil water content changes induced by high precipitation and were likely to affect the functional traits, thus shaping the significant associations of functional traits with biochemical traits. The variable responses of *S. glareosa* in different plant communities at the same site highlight the importance of species conservation and a challenge of predicting plant adaption to future climate changes. Thus, to predict plant response and adaption in the desert steppe to climate changes, we should pay more attention to the importance of integrating multi-site and multi-community studies. Moreover, community research may cover up the effects of individual species, and we should pay attention to the response mechanism of special species.

## Figures and Tables

**Figure 1 plants-09-01463-f001:**
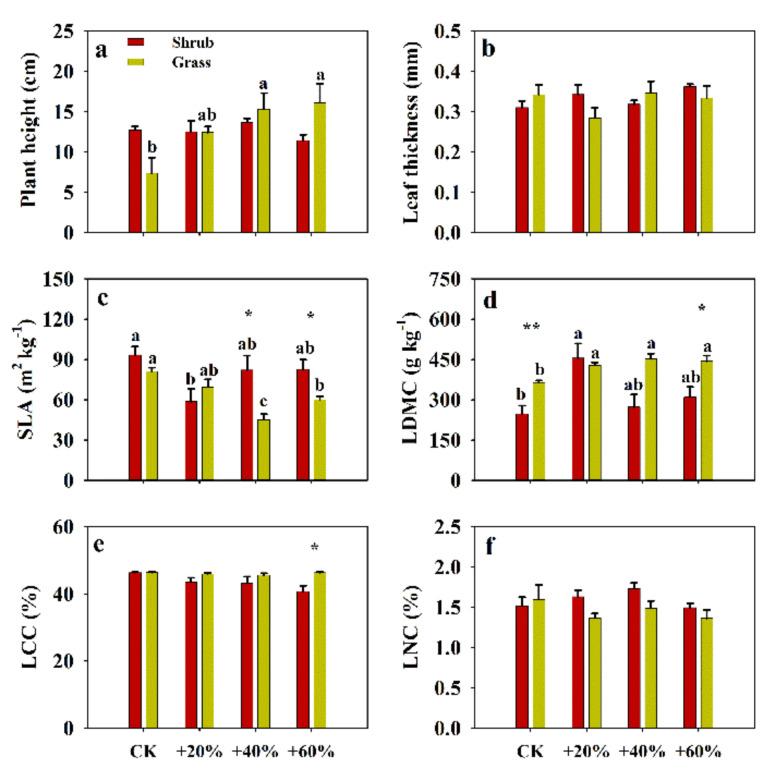
Effects of experimentally-increased precipitation on functional traits of *Stipa glareosa* in the Inner Mongolian desert steppe. (**a**) Plant height, (**b**) leaf thickness, (**c**) specific leaf area (SLA), (**d**) leaf dry matter content (LDMC), (**e**) leaf carbon content (LCC) and (**f**) leaf nitrogen content (LNC). Different lower-case letters indicate significant differences (*p* < 0.05) between treatments for the same vegetation type. Asterisks represent significant differences between different vegetation types, *: *p* < 0.05, **: *p* < 0.01.

**Figure 2 plants-09-01463-f002:**
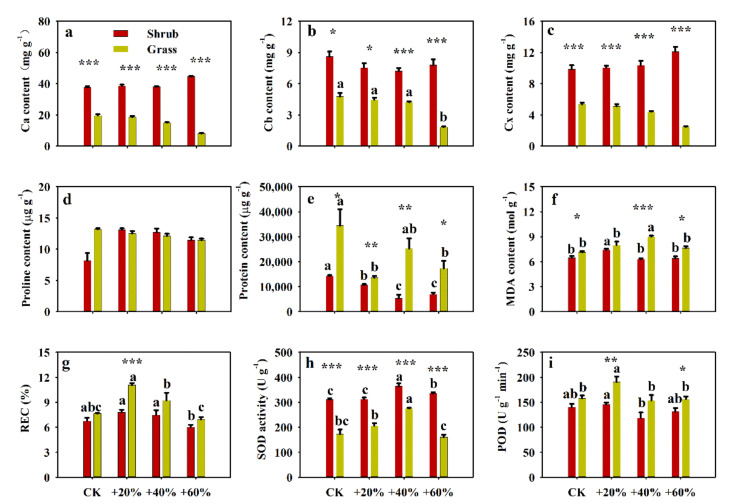
Effects of experimentally-increased precipitation on biochemical traits of *Stipa glareosa* in the Inner Mongolian desert steppe. (**a**) Chlorophyll a (Ca) content, (**b**) chlorophyll b (Cb) content, (**c**) carotenoids (Cx) content, (**d**) proline content, (**e**) protein content, (**f**) malondialaenyde (MDA) content, (**g**) relative electrical conductivity (REC), (**h**) superoxide dismutase (SOD) activity and (**i**) peroxidase (POD) activity. Different lower-case letters indicate significant differences (*p* < 0.05) between treatments for the same vegetation type. Asterisk represents significant differences between different vegetation types, *: *p* < 0.05, **: *p* < 0.01, ***: *p* < 0.001.

**Figure 3 plants-09-01463-f003:**
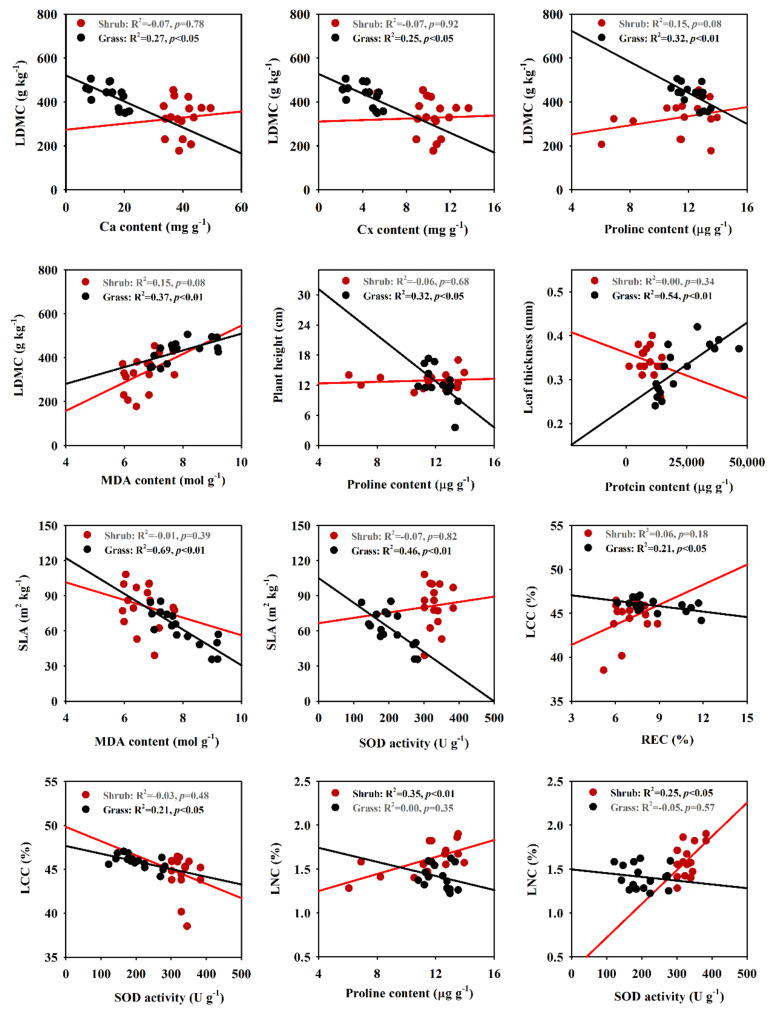
Correlations between functional and biochemical traits of *Stipa glareosa* in the shrub- and grass-dominated communities. LDMC, leaf dry matter content; Ca, chlorophyll a; Cx, carotenoids; MDA, malondialaenyde; SLA, specific leaf area; SOD, superoxide dismutase; LCC, leaf carbon content; REC, relative electrical conductivity; LNC, leaf nitrogen content.

**Figure 4 plants-09-01463-f004:**
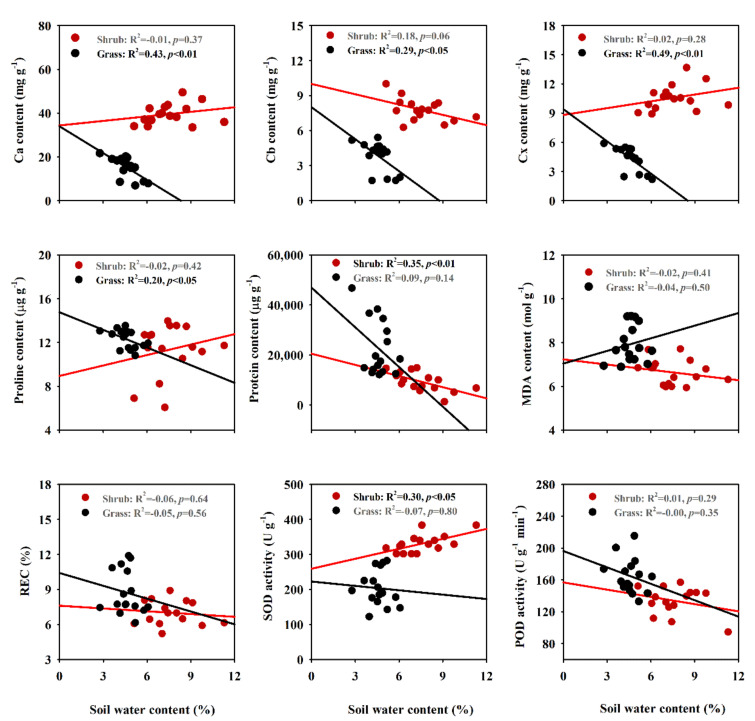
Correlations between soil water content and biochemical traits of *Stipa glareosa* in the shrub- and grass-dominated communities. Ca, chlorophyll a; Cb, chlorophyll b; Cx, carotenoids; MDA, malondialaenyde; REC, relative electrical conductivity; SOD, superoxide dismutase; POD, peroxidase.

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
