# Peer review of "Increased Precipitation Shapes Relationship between Biochemical and Functional Traits of Stipa glareosa in Grass-Dominated Rather than Shrub-Dominated Community in a Desert Steppe"

_plants, 2020, doi:10.3390/plants9111463_

Round 1
Reviewer 1 Report
This MS presents the results of a field experiment assessing the relationships between leaf functional and physiological traits in a steppe grass growing in shrub- and grass-dominated steppe. The MS has some potential to be a solid contribution, but needs a lot of very basic work to get in shape before it could be considered ready for publication here, or in any other peer-reviewed journal. In many ways, I felt as if I were reading a first draft, rather than a finished product - which makes assessing its quality problematic.
- The hypotheses presented have little or nothing to do with the introductory material. A case needs to be built in the material preceding the hypotheses that shows why should we expect different responses of this species between the two general plant community types. In addition, to the first hypothesis isn't explicitly tested, since the experimental structure is conditional (shrubland vs grassland), and does not assess variation in shrub cover on the physiological responses measured, which is what the hypothesis is addressing. To fully address this, one would need to experimentally reduce shrub cover, or isolate shrub effects and note how much positive or negative response in the grass results.
- What is the justification for selecting physiological and functional traits? What reason guided relating one parameter to another? This is a lot of information, but there is no coherent framework it is being fitted to - which turns the information into a jumble of noise. Taking the approach of regressing everything against everything and seeing what rises up to the level of significance is not robust way to do science for this very reason.
- I need more quantitative data on the structural differences between shrub and grassland community plots. All I know is that some are grass dominated and some are shrub dominated, which isn't enough.
- There are some decided multi-colinearity effects at play here. So, it is not surprising there are similar linear relationships with leaf traits (e.g. Chla and Chlb are going to co-vary, both with each other and with tissue density). This is also true of SLA and LMDC, which in many ways are two different ways at looking at one aspect of leaf function. It is not clear to me why SLA and LMDC were selected - a clearer and stronger justification of why these were chosen needs to be made.
- Speaking of leaf traits, why are SLA and LMDC the dependent variables regressed against pigment/enzyme parameters? Pigment pools and ratios and enzyme activity/pool sizes much more dynamic, and hence more likely depend on SLA and LMDC more than SLA and LMDC depend on them.
- Opening paragraph of discussion the authors imply that grass-dominated steppe is ... light limited? How do you know? This would be true for tallgrass prairie systems, but steppe? Maybe for mosses and such on the soil surface under shrubs and grass bunches, but for established grasses? Looks to me like it is just water limited.
- Finally, I have an issue with the title. These "relationships" aren't being induced in the true physiological sense of the word. Induction would require following and quantifying changes from one state to another (like induction of photosynthesis going from dark to light, where you would measure in the dark, and after the lights come on).
Reviewer 2 Report
References must be uniform - check again.
Results - are some mistakes - see attachment
Discussion - must be better, deeper explained, they are not very well organized - they should respect the order of the hypotheses
Conclusions - repeat shortly the results, from results and discussion should arise something more comprehensive and usefull
e.g. perhaps shrub communities represent a refuge for S. g. in a changing climate, as long as increased precipitation altered the grassland communities actually the dominant species. Perhaps this is available for other specialists species and can be the target for future studies and an instrumentfor species conservation

Round 2
Reviewer 2 Report
It is the second review! No comment!
Author Response
Dear Reviewer,
We appreciate for your warm work earnestly. We are pleased to note the favorable comments of you. Once again, thank you very much for your generous help.